

# Levels of PIVKA-II and alpha-fetoprotein in unresectable hepatocellular carcinoma compared to healthy controls and predictive values of both markers with radiological responses after loco-regional interventions

Norhanan Hamzah[1], Nur Karyatee Kassim[1,2,3], Julia Omar[1,3], Mohd Shafie Abdullah[3,4] and Yeong Yeh Lee[3,5]

[1] Department of Chemical Pathology, School of Medical Sciences, Health Campus, Universiti Sains Malaysia, Kota Bharu, Kelantan, Malaysia

[2] School of Dental Sciences, Health Campus, Universiti Sains Malaysia, Kota Bharu, Kelantan, Malaysia

[3] Hospital Universiti Sains Malaysia, Kota Bharu, Kelantan, Malaysia

[4] Department of Radiology, School of Medical Sciences, Health Campus, Universiti Sains Malaysia, Kota Bharu, Kelantan, Malaysia

[5] Department of Medicine, School of Medical Sciences, Health Campus, Universiti Sains Malaysia, Kota Bharu, Kelantan, Malaysia

## ABSTRACT

**Background**. The significance of the current study was to determine normative levels of PIVKA-II and AFP in patients with unresectable HCC and healthy participants. The second goal was to assess the roles of PIVKA-II and AFP in predicting radiological response after loco-regional therapy.

**Methods**. This prospective cohort study enrolled consecutive samples of HCC patients and healthy controls. Venous blood samples were obtained at baseline and after interventions to determine serum levels of PIVKA-II and AFP using the chemiluminescent microparticle immunoassay method. Radiologic responses were determined based on the WHO criteria.

**Results**. Fifty-four HCC patients (mean age 58.9 years, 49 males) and 40 healthy controls (mean age 33.5 years, 26 males) were recruited. The median serum levels of PIVKA-II and AFP in HCC *vs.* healthy controls were 988.4 *vs.* 24.2 mAU/ml and 13.6 *vs.* 1.7 ng/ml, respectively (both $p < 0.001$). With ROC curve analysis, the area under the curve (AUC) for PIVKA-II was 0.95 95% CI [0.90–0.99], and for AFP it was 0.98, 95% CI [0.95–1.0]). The cut-off value for PIVKA-II was 41.4 mAU/ml, and AFP was 4.8 ng/ml. PIVKA-II levels correlated significantly with radiological responses ($r = 0.64$, $p = 0.02$) but not AFP ($r = 0.09$, $p = 0.2$).

**Conclusion**. PIVKA-II and AFP levels are distinctive between unresectable HCC and healthy controls. However, PIVKA-II, not AFP, can predict the radiological response after loco-regional therapy.

Corresponding author
Nur Karyatee Kassim,
karyatee@usm.my

## INTRODUCTION

Hepatocellular carcinoma (HCC) is the sixth most common malignancy globally (*Bray et al., 2018*), with Asian populations accounting for approximately 70% of all cases (*Torre et al., 2015*). HCC is the eighth most prevalent cancer in Malaysia and the sixth most common cancer in men (*Omar & Ibrahim Tamin, 2011*). Unresectable liver cancer, or advanced-stage HCC, refers to tumours that cannot be surgically removed due to factors such as size, location, involvement of blood vessels, or metastasis (*Cheng et al., 2012*; *Llovet et al., 2018*). HCC was discovered at an advanced stage in about two-thirds of cases, with a median survival time of only four months (*Ferlay et al., 2015*; *Khalaf et al., 2017*). The management of unresectable liver cancer involves several treatment modalities, including loco-regional (for example, trans-arterial chemoembolization (TACE) and radioembolization) and systemic therapy (for example, sorafenib) (*Llovet et al., 2018*).

Radiological imaging, especially computed tomography (CT), is commonly used to diagnose and evaluate tumour response after treatment but is limited by poor differentiation of viable tumours from necrotic or fibrotic tissue (*Arora & Kumar, 2014*). Pathological remission can occur without evidence of radiological response (*Llovet et al., 2018*), most likely due to desmoplastic and inflammatory reactions caused by related ischemia and tissue edema that developed following a specific treatment (*Arora & Kumar, 2014*). To address this limitation, serum tumour markers such as alpha-fetoprotein (AFP) or Protein Induced by Vitamin-K Absence-II (PIVKA-II) may be more convenient, non-invasive, repeatable, and inexpensive.

AFP is a glycoprotein produced by the yolk sac in the early stages of development and later by the liver. AFP is a biomarker already widely used for diagnosis and may be beneficial in predicting treatment outcomes (*Lau et al., 2001*; *Kim et al., 2011*; *Song et al., 2017*; *Yu, 2016*; *Kondo, Kimura & Shimosegawa, 2015*; *Hatzaras et al., 2014*; *Zhang et al., 2018*; *Plano Sánchez et al., 2018*). However, AFP has a reportedly low sensitivity at a cut-off level of 20 ng/ml (*Gupta, Bent & Kohlwes, 2003*). Furthermore, AFP can be normal in up to 35% of small HCCs and may be elevated in other benign liver disorders, such as hepatitis, cirrhosis without HCC, and inactive hepatocyte regeneration, in addition to being influenced by age and sex (*Yu et al., 2016*; *AlSalloom, 2016*).

PIVKA-II, also known as Des-$\gamma$-Carboxy-Prothrombin (DCP), is abnormal prothrombin that lacks coagulation activity. In the presence of HCC, PIVKA-II levels will be elevated due to decreased activity of gamma-glutamyl carboxylase and low vitamin K levels in the cancer tissue (*Xing et al., 2016*). PIVKA-II levels do not always correlate with AFP levels (*Xing et al., 2018*; *Park et al., 2012*). Whereas AFP levels reflect intrahepatic tumour burden, PIVKA-II levels reflect tumour behaviour such as vascular invasion and extrahepatic disease (*Park & Park, 2013*; *Ji et al., 2016*; *Kang et al., 2012*).

PIVKA-II responses post hepatectomy might be valuable in the early detection of tumour recurrence (*Nanashima et al., 2011*; *Kim et al., 2007*).

It is unclear whether PIVKA-II can completely replace or enhance the role of AFP in HCC diagnosis among different populations. Furthermore, the correlations between AFP, PIVKA-II, and radiological responses following loco-regional treatment have not been thoroughly explored.

Recent research has investigated PIVKA-II and AFP as potential predictive markers for unresectable liver cancer (*Park & Park, 2013*; *Feng et al., 2021*). Studies have indicated that elevated levels of PIVKA-II are associated with tumour aggressiveness, tumour recurrence, and poor prognosis in unresectable liver cancer patients (*Feng et al., 2021*). Research has demonstrated that higher AFP levels are correlated with larger tumour size, vascular invasion, and advanced-stage disease in patients with unresectable liver cancer (*Llovet et al., 2018*). It is important to note that relying solely on single biomarkers may not provide sufficient accuracy or reliability for predicting outcomes in unresectable liver cancer (*Galle et al., 2018*). Therefore, identifying complementary biomarkers or developing multi-marker panels is crucial to improve predictive accuracy.

Hence, our study was conducted to determine PIVKA-II and AFP's normative levels and diagnostic performance in healthy controls *vs.* patients with unresectable HCC. In addition, we investigated the correlations between PIVKA-II and AFP levels with the radiological response following loco-regional interventions to understand the potential use of these biomarkers in clinical practice.

## STUDY POPULATIONS & METHODS

### Study populations

This prospective cohort study involved consecutive samples of patients diagnosed with unresectable HCC. The research was conducted at Hospital Universiti Sains Malaysia (USM), a tertiary university hospital serving the northeastern region of Peninsular Malaysia. Consecutive healthy volunteers were enrolled through advertisements. Volunteers included hospital staff, students, family members, and accompanying persons.

Inclusion criteria were adult patients above 18 years old, of either gender, and a confirmed diagnosis of unresectable HCC based upon typical imaging features on CT and subsequently treated with loco-regional therapies based on clinician discretion. Treatment options included transarterial chemoembolization (TACE), radiofrequency ablation (RFA), microwave ablation, percutaneous ethanol injection (PEI), or systemic therapy (*e.g.*, sorafenib). The typical CT findings of HCC are enhancement during the arterial phase and early washout in the portal phase. In addition, tumour number, tumour diameter (longest axis of the largest tumour grouped into <3 cm, 3–5 cm, and >5 cm) (*Kim et al., 2007*), and presence or absence of portal vein thrombosis were recorded. Healthy participants were adult volunteers above 18 years of age, of either gender, and did not have any history of chronic medical or surgical illnesses, no history of alcohol consumption, no family history of HCC, and normal liver biochemistry. Exclusion criteria were patients on warfarin or vitamin K within six months of enrolment, hepatic tumours other than

HCC, liver metastases, previous liver surgeries, neo-adjuvant therapy, and currently on chemotherapy.

The study protocol was reviewed and approved by the Human Research Ethics Committee of USM (USM/JEPeM/18010058) and was conducted in accordance with the ethical standards of the 1964 Helsinki Declaration. All subjects were over 18 years of age and voluntarily signed the informed consent forms.

## Determination of PIVKA-II and AFP levels

Serum PIVKA-II and AFP levels were measured in all patients at baseline and six weeks following treatment. Only baseline PIVKA-II and AFP values were collected from healthy controls. Both serum PIVKA-II and AFP levels were determined using the chemiluminescent microparticle immunoassay (CMIA) (ARCHITECT Plus analyzer, Abbott, Wiesbaden, Germany) performed according to the manufacturer's instructions. According to the manufacturer's insert kit, the verified reference interval is 11.12–32.01 mAU/ml for PIVKA-II and 10 ng/ml for AFP. A clinical response to treatment was defined as a reduction in PIVKA-II and AFP levels of more than 50% from baseline (*Park & Park, 2013*).

## Evaluation of radiological response

Radiologic tumour response was evaluated by CT scans repeated six weeks after treatment and assessed using the WHO criteria (*Eisenhauer et al., 2009*). Tumour responses were divided into four categories: complete response (CR), partial response (PR), progressive disease (PD), and stable disease (SD). CR was defined as complete disappearance of all lesions; PR was defined as a 50% or more significant decrease in the sum of all areas (longest diameters multiplied by longest perpendicular diameter); PD was defined as a more than 25% increase in the product of two perpendicular diameters of the largest tumour nodule, or one of the measurable lesions, or the appearance of new lesions. All other findings were grouped as stable disease (SD).

## Statistical analysis

Numerical data were expressed as mean (standard deviation) (SD) if not otherwise mentioned. Categorical data were expressed as frequency and percentages. Receiver operating characteristic (ROC) curve analysis was performed using the IBM SPSS Statistics software Version 26.0 (SPSS Inc., Chicago, IL, USA) to obtain the area under the curve (AUC), cut-off values, sensitivity, specificity, positive predictive values (PPV) and negative predictive values (NPV) of PIVKA-II and AFP. Mann–Whitney U was used to calculate and compare AUC. The Fisher's exact or chi-square test was performed to determine the association of PIVKA-II and AFP levels with radiological responses. Non-parametric Spearman's rank correlation ($r_s$) was employed to correlate PIVKA-II and AFP serum levels with radiological responses in HCC. A $p$-value of <0.05 was considered statistically significant.

## RESULTS

### Normative PIVKA II and AFP levels

Fifty-four HCC patients and 40 healthy controls were sequentially enrolled in this study. Out of 125 HCC patients screened for eligibility, 54 were eventually enrolled, 12 died before completion of the study, 22 patients defaulted subsequent follow-ups, and 20 patients completed pre- and post-treatment blood tests. Similarly, 50 controls were screened in a sequential manner, and 40 satisfied the eligibility. The demography of the study participants is shown in Table 1. For the HCC group, the mean (SD) age was 58.9 (9.3) years with a male preponderance ($n = 49$, 90.7%), and for the healthy controls, the mean (SD) age was 33.5 (10.3) years. Chronic hepatitis B (HBV) was the most common cause of HCC ($n = 29$, 53.7%), followed by chronic hepatitis C (HCV) ($n = 9$, 16.6%) and non-alcoholic fatty liver disease (NAFLD) ($n = 9$, 16.6%). Of the 54 HCC patients, half (59.3%, $n = 32$) opted for conservative management, while 14.8% or $n = 8$ underwent TACE or microwave ablation, 9.2% or $n = 5$ underwent PEI, and only 1 (1.9%) patient underwent RFA. More than half (64.8% or $n = 35$) had multiple nodules, and 21 (38.9%) patients had tumour sizes of more than five cm at baseline.

PIVKA-II serum levels were substantially greater in HCC than in the healthy group with a median (IQR) value of 988.4 (23832.8) mAU/ml *vs.* 24.2 (10.5) mAU/ml, ($p = 0.001$). Similarly, the median (IQR) level of AFP at baseline was significantly higher in HCC compared to the healthy group [13.6 (647.83) ng/ml *vs.* 1.7 (1.21) ng/ml, $p = 0.001$] (Table 2).

### ROC curve analysis

As shown in Fig. 1, both PIVKA-II and AFP could distinguish HCC from healthy controls. PIVKA-II was observed to have relatively similar AUCs with AFP (AUC PIVKA-II = 0.95 95% CI [0.90–0.99]; AUC AFP = 0.98, 95% CI [0.95–1.0]). When PIVKA-II and AFP were combined, the diagnostic power improved significantly compared to AFP or PIVKA-II (AUC PIVKA-II = 0.99, 95%CI [0.97–1.00]) ($P < 0.05$).

The optimal cut-off value for PIVKA-II was 41.4mAU/ml with 87.5 percent sensitivity, 100 percent specificity, 97.9 percent PPV, and 84.8 percent NPV. For AFP, the optimal cut-off value was 4.85ng/ml with a sensitivity of 90.7 percent, specificity of 100 percent, PPV of 100 percent, and NPV of 88.9 percent. The combination of these tumour markers yielded a sensitivity and specificity of 98% and 100%, respectively.

### Association of PIVKA II and AFP levels with radiological response

Of 20 patients assessed with PIVKA-II, eight (40%) were clinical responders, while twelve (60%) were non-responders. When based on the WHO criteria, of clinical responders, two (10%) had CR, six (30%) had PR, eight (40%) had PD, and four (20%) had SD. Seven (35%) of the 20 patients evaluated with AFP were clinical responders, while thirteen (65%) were non-responders. Based on the WHO criteria, of clinical responders with AFP, two (10%) patients had CR, six (30%) had PR, eight (40%) had PD, and four (20%) had SD.

**Table 1  Sociodemographic and clinical characteristics of study populations.**

|  | HCC, $n = 54$ n (%) | Healthy, $n = 40$ n (%) |
|---|---|---|
| Age, y | 58.9 (9.28)[*] | 33.5 (10.32)[*] |
| Gender |  |  |
| Male | 49 (90.7%) | 26 (65%) |
| Female | 5 (9.3%) | 14 (35%) |
| Ethnics |  |  |
| Malay | 49 (90.7%) |  |
| Non Malay | 5 (9.3%) |  |
| Risk factor |  |  |
| Infective risk |  |  |
| HBV infection | 29 (53.7%) |  |
| HCV infection | 9 (16.6%) |  |
| HBV and HCV co-infection | 5 (9.3%) |  |
| Non-infective risk |  |  |
| NAFLD | 9 (16.6%) |  |
| AIH | 1 (1.9%) |  |
| Alcoholic liver disease | 1 (1.9%) |  |
| Tumour number (nodule) |  |  |
| Single | 19 (35.2%) |  |
| Multiple | 35 (64.8%) |  |
| Tumour size |  |  |
| <3cm | 19 (35.2%) |  |
| 3–5 cm | 14 (25.9%) |  |
| >5 cm | 21 (38.9%) |  |
| Portal vein thrombosis |  |  |
| Absent | 40 (74.1%) |  |
| Present | 14 (25.9%) |  |
| Treatment options |  |  |
| TACE | 8 (14.8%) |  |
| RFA | 1 (1.9%) |  |
| PEI | 5 (9.2%) |  |
| Microwave ablation | 8 (14.8%) |  |
| Conservative | 32 (59.3%) |  |

Notes.
HBV, hepatitis B virus; HCV, hepatitis C virus; NAFLD, Non-alcoholic Fatty liver disease; AIH, autoimmune hepatitis; TACE, transarterial chemoembolization; RFA, radiofrequency ablation; PEI, percutaneous ethanol injection.
*mean (SD)

Results of Spearman's rank correlation ($rs$) test between PIVKA-II, AFP, and radiological responses in HCC are shown in Table 3. PIVKA-II was strongly correlated with radiological responses ($p = 0.016$, $r = 0.64$) but not AFP ($p = 0.1873$, $r = 0.09$).

**Table 2  PIVKA -II and AFP level among HCC patients and healthy populations.**

| Serum markers at baseline | HCC, $n = 54$ (Median (IQR) | Healthy, $n = 40$ (Median (IQR) | *p*-value |
|---|---|---|---|
| PIVKA-II level (mAU/ml) | 988.4 (23832.82) | 24.2 (10.55) | <0.001 |
| AFP level (ng/ml) | 13.6 (647.83) | 1.7 (1.21) | <0.001 |

**Notes.**
PIVKA-II, Protein Induced by Vitamin-K Absence-II; AFP, alpha fetoprotein.
Statistical test: Mann–Whitney test for comparison between two groups.

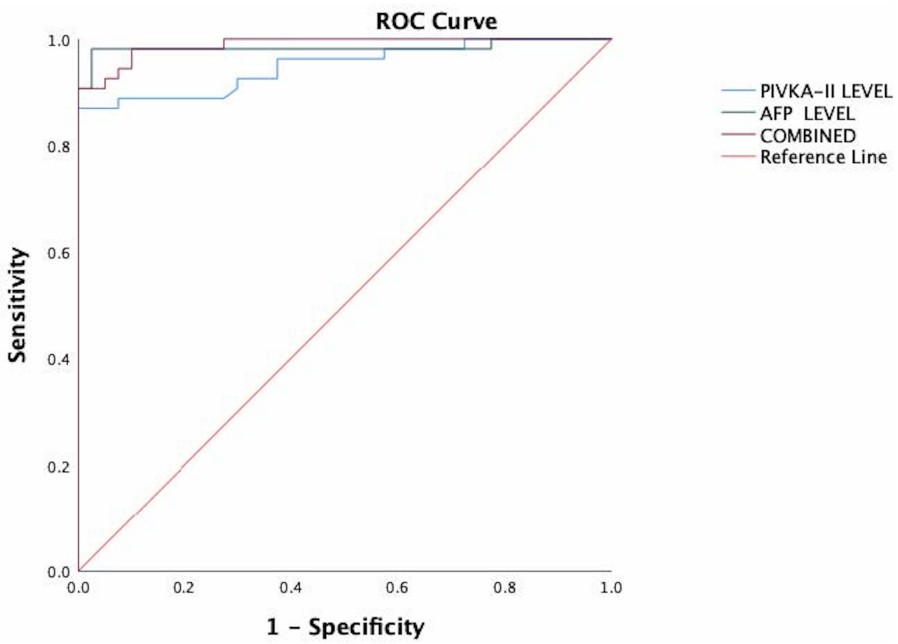

**Figure 1  The ROC of PIVKA-II and AFP for the diagnosis of hepatocellular carcinoma in all patients.**
PIVKA-II has similar AUCs with AFP (AUC PIVKA-II = 0.95 95% CI [0.90–0.99]; AUC AFP = 0.98, 95% CI [0.95–1.0]). When PIVKA-II and AFP were combined, the diagnostic power improved significantly compared to either AFP or PIVKA-II (AUC PIVKA-II = 0.99, 95% CI [0.97–1.00]) ($P < 0.05$).

## DISCUSSION

A summary of notable findings is as follows. First, both PIVKA-II and AFP levels are significantly elevated in unresectable HCC, with clear distinctive levels from healthy controls, similarly reported elsewhere (*Ette et al., 2015*). A previous study has also reported an assessment of biomarkers between patient and control to avoid misleading interpretation (*Rutjes et al., 2005*; *Husnain et al., 2021*). Second, PIVKA-II and AFP have relatively similar AUC values, sensitivity, specificity, PPV, and NPV, and the combination of both markers yielded higher sensitivity and specificity. Thirdly, PIVKA-II levels correlated with radiological responses post-locoregional interventions but not AFP levels.

The highly distinctive values of PIVKA-II and AFP in HCC *vs.* controls might be partly explained by half of HCCs being caused by HBV infection. Chronic hepatitis B is highly prevalent in Malaysia, consistent with epidemiological studies of other Asian populations

**Table 3    Association between serological response and radiological response among HCC patients.**

|  | Radiologic response (WHO criteria) | | | | $r_s$[b] | *p*-value |
|  | Complete response | Partial response | Stable disease | Progressive disease | | |
|---|---|---|---|---|---|---|
| PIVKA-II response |  |  |  |  | 0.64 | 0.016[a] |
| PIVKA-II responder ($n = 8$, 40%) | 1 (12.5%) | 5 (62.5%) | 2 (25%) | 0 (0) |  |  |
| PIVKA-II non-responder ($n = 12$, 60%) | 1 (8.3%) | 1 (8.3%) | 2 (16.7%) | 8 (66.7%) |  |  |
| AFP response |  |  |  |  | 0.09 | 0.187 |
| AFP responder ($n = 7$, 35%) | 2 (28.6%) | 1 (14.3%) | 1 (14.3%) | 3 (42.9%) |  |  |
| AFP non-responder ($n = 13$, 65%) | 0 (0) | 5 (38.9%) | 3 (23.1%) | 5 (38.5%) |  |  |

**Notes.**

Results are expressed as n (%), *PIVKA-II* Protein Induced by Vitamin-K Absence-II.

[a]Fisher's exact test was applied.

[b]Spearman's rank correlation ($r_s$).

(*Zakhary et al., 2013*; *Raihan, 2016*; *Yu et al., 2015*; *Ng & Wu, 2012*). Tumour markers have greater levels due to larger bilobar masses and aggressive behavior associated with chronic HBV infection (*Raihan, 2016*). PIVKA-II is a newer marker than AFP and a potentially better marker for HCC (*Zhang et al., 2014*; *Seo et al., 2015*; *Chon et al., 2012*). According to one study that compared PIVKA-II and AFP values among HBV-related HCC, PIVKA-II is the better marker, and their combination may improve early HCC detection (*Ng & Wu, 2012*; *Chon et al., 2012*).

Based on the ROC analysis, PIVKA-II had relatively similar AUCs as AFP and sensitivity, specificity, PPV, and NPV. These findings were consistent with previous studies (*Chon et al., 2012*; *Hyoung et al., 2009*; *Xing et al., 2016*). However, the exact sensitivity and specificity values of PIVKA-II were different from other studies, which could be attributed to varying sample sizes and different study designs (*Chon et al., 2012*; *Nakamura et al., 2006*). A combination of PIVKA-II and AFP yielded better sensitivity and specificity, and this would be important considering that AFP may be normal in a third of HCCs (*Galle et al., 2018*).

After loco-regional treatment in unresectable HCCs, imaging is considered the gold standard in assessing treatment response and subsequent treatment strategies. However, loco-regional interventions may alter imaging characteristics and size determination of target lesions because of intra-tumour edema, hemorrhage, or necrosis (*Marin et al., 2015*). There is also a potential cancer risk from repeated exposure to radiation with CT scans (*Hennedige & Venkatesh, 2012*). Furthermore, radiological characteristics after TACE can become non-homogenous and inconsistent in some cases due to irregular uptake of lipiodol and liquefaction necrosis (*Hennedige & Venkatesh, 2012*). In summary, it can be challenging to distinguish clinical responses based on tumour appearance on CT scans after loco-regional therapy, whether due to post-treatment changes, residual lesions, or recurrent disease (*Marin et al., 2015*). Therefore, tumour marker evaluation post-treatment may be more objective, easier to measure, and relatively inexpensive compared to imaging (*Arai et al., 2014*).

The current study found a significant correlation between PIVKA-II levels and radiological response post-intervention. This outcome was consistent with a few published

studies in Asia. For example, *Arai et al. (2014)* concluded that PIVKA-II trends were strongly associated with overall response and disease-free rates in patients with recurrent HCC treated with TACE (*Hennedige & Venkatesh, 2012*). Similarly, *Park et al. (2014)* observed that the PIVKA-II response was associated with radiological response and was predictive of tumour progression as well as overall survival in HCC patients undergoing TACE.

In contrast, no significant association was discovered between AFP levels and radiological responses following the intervention. The possible explanation may be due to AFP levels that did not normalize completely, although the tumour had been eradicated (*Arai et al., 2014*). In a similar study, *Park et al. (2014)* explored the role of PIVKA-II and AFP in predicting non-surgical treatment outcomes in advanced HCC and found that a combination of biomarkers predicted tumour responses to local treatments better than AFP alone. In another study, AFP serum levels were significantly correlated with the radiological responses post-TACE, but PIVKA-II serum levels were not. There are discrepancies between studies, and further extensive studies may be warranted.

There are several limitations in our study. First, it was conducted in a single center over one year period during the COVID-19 pandemic. Only about half of the subjects returned for a repeat CT scan and blood investigations, resulting in a limited sample size. However, a similar study by *Feng et al. (2021)*; found significant results with the same limitations (*Galle et al., 2018*). Second, in the current study, the age and sex of patients and controls were relatively mismatched due to the nature of HCC being more common in older age groups and among males. The disparity in age ranges between patients and controls arose as HCC is expectedly more common in the older age groups (*El-Serag & Rudolph, 2007*).

Moreover, obtaining healthy elderly control is difficult. Pertaining to male dominance, a previous study has also shown similar findings (*Gomaa et al., 2008*). In addition, comparing HCC with controls may inflate the diagnostic performance of the test. Third, we did not compare levels of both markers across different aetiologies (viral hepatitis *vs.* NAFLD) or stages of disease (fibrosis *vs.* cirrhosis).

In conclusion, PIVKA-II and AFP levels in unresectable HCC patients are significantly higher and distinctive from healthy controls. PIVKA-II may be more promising than AFP in predicting radiological response after loco-regional interventions.

### Funding

This research was funded by Universiti Sains Malaysia Short-Term Research Grant (304.PPSG.6315413). The funders had no role in study design, data collection and analysis, decision to publish, or preparation of the manuscript.

### Grant Disclosures

The following grant information was disclosed by the authors:
Universiti Sains Malaysia Short-Term Research Grant: 304.PPSG.6315413.

## Competing Interests

The authors declare there are no competing interests.

## Author Contributions

- Norhanan Hamzah conceived and designed the experiments, performed the experiments, analyzed the data, prepared figures and/or tables, authored or reviewed drafts of the article, and approved the final draft.
- Nur Karyatee Kassim conceived and designed the experiments, performed the experiments, analyzed the data, prepared figures and/or tables, authored or reviewed drafts of the article, and approved the final draft.
- Julia Omar analyzed the data, prepared figures and/or tables, authored or reviewed drafts of the article, and approved the final draft.
- Mohd Shafie Abdullah conceived and designed the experiments, analyzed the data, authored or reviewed drafts of the article, and approved the final draft.
- Yeong Yeh Lee conceived and designed the experiments, authored or reviewed drafts of the article, and approved the final draft.

## Human Ethics

The following information was supplied relating to ethical approvals (*i.e.*, approving body and any reference numbers):

Human Research Ethics Committee of Universiti Sains Malaysia (USM/-JEPeM/18010058)

## Data Availability

The raw data of measurement of the PIVKA ii levels of HCC patients and healthy subjects are available in the Supplementary File.

## Supplemental Information

Supplemental information for this article can be found online at http://dx.doi.org/10.7717/peerj.15988#supplemental-information.

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
