# Peer review of "Levels of PIVKA-II and alpha-fetoprotein in unresectable hepatocellular carcinoma compared to healthy controls and predictive values of both markers with radiological responses after loco-regional interventions"

_PeerJ, doi:10.7717/peerj.15988_

## Round 0.1 · original submission · Major Revisions

Major concerns emerged from the reviews. Please revise the manuscript according to the suggestions received.

·

Basic reporting

It is a good paper sufficiently clear

Experimental design

Some concern is present:
authors compare HCC patients with healthy control and this can artificially enflate diagnostic performance. This should bve discussed

Methods should be better spercified: Which method was used to calculaste and compare AUC

Validity of the findings

See first comment in the above section

·

Basic reporting

The article is overall well written except some sentences, for example, in the beginning of the "RESULTS ", it's better descript the sample enrolled methods brifly instead of "of those screened" in line 149. Literature references, sufficient field background/context provided.

Experimental design

Hamzah et al tried to determine the dependency of the levels of serum levels of PIVKA-II and AFP in patients with unresectable HCC and healthy participants. The authors gathered 54/125 HCC patients and 40/50 controls and determination the levels using chemiluminescent microparticle immunoassay(CMIA) assays.

Validity of the findings

The current research shows important significance but a bit simple in structure overall.
1. There should better be the same interval range seting between the enrolled HCC patients and the healthy control groups for obtaining credible data or results to eliminate unnecessary errors and factors, for example, the mean(SD) age of the control group 33.5 years which is more smaller than the HCC group 58.9. Moreover, the ratio of the gender also shows a big difference.
2. Are the levels of PIVKA-II showed correlation with AFP values within the HCC patients? Moreover, is it correlation of PIVKA-II with HBV or HCV caused HCC?
3. For the association of PIVKA II and AFP levels with radiological response, the sample size is too small to obtain convincing results. It’s better to expand each sample size of the CR, PR, PD or SD groups.

Reviewer 3 ·

Basic reporting

The author analyzed the differences in two biomarkers, PIVKA-II and AFP, between patients with unresectable liver cancer and healthy individuals. They examined the diagnostic value of these biomarkers as well as their predictive value for radiotherapy response. Although the study design was relatively comprehensive, the content of the article was overly simplistic and one-dimensional, and there were some issues with language and writing. Therefore, further significant revisions are still needed by the author.
For example, the background section of the abstract does not indicate the significance or value of the study, nor does it state the research purpose.

Experimental design

The introduction section only emphasizes hepatocellular carcinoma (HCC) without providing detailed information on the main branch of the study, which is unresectable liver cancer. It also fails to highlight the current issues related to this disease or clarify the research’s value.
Diagnostic studies generally consist of three stages, namely, the discovery stage, training stage, and validation stage. In this article, the author draws conclusions based on a very small sample size, which weakens the reliability. It is recommended to increase the sample size to approach validation or include external validation using TCGA or GEO datasets.

Validity of the findings

The statistical description section should only include the statistical methods used in the article and how they were chosen. Methods that were not utilized, such as Kruskal-Wallis, should not be mentioned. The Kruskal-Wallis test is widely known as a non-parametric test for comparing medians among three or more groups, this article does not involve comparisons among multiple groups.
There are still several writing errors throughout the article, such as the sudden appearance of “)” in line 43 and the use of “.” before the reference list in lines 57 and 59. Overall, the author should ensure consistency and readability, as this reflects the author’s attitude. It is recommended to carefully revise the manuscript.

Additional comments

no comment

---

## Round 0.2 · accepted · Accept

The authors have addressed all the comments from reviewers properly. The manuscript is ready for publication.

·

Basic reporting

The authors have addressed my previous coments in this part.

Experimental design

The authors have addressed my previous coments in this part.

Validity of the findings

The authors have addressed my previous coments in this part.

Additional comments

none